# Modeling the Effects of Threading Dislocations on Current in AlGaN/GaN HEMT

**DOI:** 10.3390/mi14020305

**Published:** 2023-01-24

**Authors:** Censong Liu, Jie Wang, Zhanfei Chen, Jun Liu, Jiangtao Su

**Affiliations:** Zhejiang Key Laboratory of Large-Scale Integrated Circuit Design, Hangzhou Dianzi University, Hangzhou 310018, China

**Keywords:** AlGaN/GaN HEMTs, threading dislocations, current collapse, gate leakage current, modeling

## Abstract

The aim of this paper is to model the effects of threading dislocations on both gate and drain currents of AlGaN/GaN high electron mobility transistors (HEMTs). The fraction of filled traps increases with the threading dislocations, while the trapping effects cause a decrease in drain current and an increase in gate leakage current. To model the drain current drop, the two simplified RC subcircuits with diodes are proposed to capture the charge trapping/detrapping characteristics. The trap voltages *V_g_trap_* and *V_d_trap_* generated by RC networks are fed back into the model to capture the effects of traps on drain current. Considering acceptor-decorated dislocations, we present a novel Poole–Frenkel (PF) model to precisely describe the reverse leakage gate current, which plays a dominant role in the gate leakage current. The proposed model, which uses physical parameters only, is implemented in Verilog-A. It is in excellent agreement with the experimental data.

## 1. Introduction

Gallium nitride is the representative of the third-generation semiconductor [1,2], which has the advantages of high breakdown electric field, high thermal conductivity, high electron mobility, and high operating temperature. Due to the above-mentioned excellent properties, GaN-based HEMTs are considered to be one of the best choices for high-power [3] and high-frequency [4] applications.

However, dislocations are inevitably distributed in different regions of the GaN HEMTs due to factors such as materials and the ideality degree of the manufacturing process. According to current reports, the most typical and unavoidable dislocation in GaN and its heterostructures is threading dislocation [5,6]. Threading dislocations can cause parasitic effects such as negative charge trapping under the gate region [7], hot carrier trapping in epitaxial structures after long periods of electrical stress and threshold voltage instability [8,9]. The filling of traps affects the presence of two-dimensional electron gas (2-DEG) in the channel region, which in turn affects the drain current [10] and gate current [11]. Furthermore, since the electron trap center acts as a fixed charge to induce a tunneling path in the AlGaN barrier [12,13], the gate leakage current increases. The high gate leakage current will cause severe electrical reliability problems and result in unprecedented poor performances for the device [14,15]. For instance, it will cause the degradation of breakdown voltage [13] and increase in noise figure [16]. However, there is little literature studying the effects of threading dislocations on current in GaN HEMTs [6,17,18]. The trapping effects caused by threading dislocations are nonnegligible in building device current models. Therefore, it is in great demand to develop a model to predict the effects of threading dislocations on the current of AlGaN/GaN HEMTs.

Few reported models have considered the effects of threading dislocations on both drain current and gate current in GaN HEMTs. Threading dislocations have effects on the fraction of filled traps, while trapping effects have a great influence on drain current. There are two common methods for modeling the effects of the traps on the drain current of GaN HEMTs. One is to capture the effects of traps caused by threading dislocations dynamically through RC networks [19,20,21,22], and the other is to model the trapping effects through the trap model based on the Shockley–Read–Hall (SRH) theory [23,24]. Compared with the trap model based on the RC network, the physical trap model based on the SRH theory requires a large amount of calculation and has a slow simulation speed. However, most trap models based on RC networks do not distinguish the charging and discharging paths of traps. For instance, ASM-HEMT model has empirically modified the model parameters such as the cut-off voltage ($V_{off}$) according to trap voltages generated by the RC network, capturing the drain current drop caused by traps [19]. However, its subcircuits are unable to generate two different time constants for capture and emission processes. Although the MVSG model employs RC or diode-capacitance networks to change either the ON resistance of the device ($R_{on}$) or the threshold voltage ($V_{T}$) dynamically, it cannot generate two different time constants for charge trapping and detrapping [20].

Compared with the prosperous drain current models, there are only a few gate current models [25,26] as well as several constructive transport models. The gate current models mainly include the Poole–Frenkel (PF) emission, trap-assisted tunneling (TAT) [27], Fowler–Nordheim tunneling (FNT) [28], etc. However, these gate current models rarely consider the effects of threading dislocations on gate current in AlGaN/GaN HEMTs.

Based on the proposed models, this paper makes an improvement. We model the effects of threading dislocations on both gate current and drain current of AlGaN/GaN HEMTs. Two parallel RC trap networks with diodes are used to characterize the trapping effects on drain current. The RC subcircuits with diodes can generate two different time constants for the charge capture and emission processes. The trapping effects are modeled by modifying the model parameters such as $V_{off}$ according to trap voltages generated by RC subcircuits. Through studying the effects of threading dislocations on gate current under reverse voltage, we present a new transport mechanism, two-state PF emissions resulting from two distinct electron-transfer channels introduced by threading dislocations [15], to model the effects of threading dislocations on gate current under reverse voltage. The proposed model is applied to model a GaN HEMT device with 0.25 μm gate length, 125 μm gate width and six gate fingers. The model accuracy was demonstrated by the excellent agreement between the modeled data and measured data.

The rest of this paper is organized as follows. Section 2 elaborates on the procedure of model development and its parameter extraction. Section 3 illustrates the validation of the proposed model, and Section 4 concludes this paper.

## 2. Description and Parameter Extraction Methods

This section is composed of two parts, which illustrate the modeling of the effects of threading dislocations on drain current and gate leakage current, respectively.

### 2.1. Modeling the Effects of Threading Dislocations on Drain Current

In this part, two RC subcircuits with diodes are proposed to model the trapping effects on drain current affected by threading dislocations. The equivalent circuit topology of the trap model of drain current is shown in Figure 1. As one can see, each RC subcircuit has two resistors, where the charging and discharging paths are distinguished by diodes. The capacitances of the subcircuits are charged or discharged by the input voltages of the gate and drain terminals, and the generated trap voltages $V_{g\_trap}$ and $V_{d\_trap}$ are fed back into the model to update the $V_{off}$, mobility degradation coefficient (i.e., $\mu_{a}$) and other model parameters, capturing the trapping effects.

The drain current is calculated using the surface-potential model [29] and given as follows:(1)$$I_{ds}=\frac{W\cdot NF}{L}\cdot \frac{\mu_{eff}\cdot C_{g}}{\sqrt{1+\theta_{sat}^{2}\psi_{ds}^{2}}}\cdot \left(V_{go}-\psi_{m}+V_{th}\right)\cdot \psi_{ds}\cdot \left(1+\lambda \cdot V_{ds,eff}\right)$$
(2)$$\mu_{eff}=\frac{\mu_{0}\left(T\right)}{1+\mu_{a}\cdot E_{y,eff}+\mu_{b}\cdot E_{y,eff}^{2}}$$
(3)$$V_{go}=V_{gs,eff}-V_{off}+eta0\cdot \left(\frac{V_{dsx}\cdot V_{dscale}}{\sqrt{V_{dsx}^{2}+V_{dscale}^{2}}}\right)$$
where $\psi_{ds}=\psi_{d}-\psi_{s}$, $\psi_{m}=\left(\psi_{d}+\psi_{s}\right)/2$. $\theta_{sat}$ is the velocity saturation effect parameter and λ is the channel length modulation effect parameter [23]. The mobility degradation due to the vertical field is included in $\mu_{eff}$. A large number of real device effects such as DIBL, self-heating effects, and nonlinear access region resistance have been included in the complete model of $I_{d}$ to represent real GaN HEMTs [29].

The values of $R_{g\_capt}$, $R_{d\_capt}$, $R_{g\_emit}$, $R_{d\_emit}$, $C_{g\_trap}$, and $C_{d\_trap}$ in the trap model need to be determined according to the capture and emission time constants. The time constants for charge capture and emission are calculated as follows:(4)$$\tau_{capt,emit}=R_{capt,emit}\times C_{trap}$$

The time of charge trapping is very short, about $10^{-10}$ s, and the charge takes a long time to release, about $10^{-5}$ s [30].

Pulsed $I_{DS}-V_{DS}$ measurement is a common method to study the charge capture and emission processes of GaN HEMTs [31]. In the pulsed $I_{DS}-V_{DS}$ measurements, signal generators are applied at the drain and gate terminals to provide inputs, as shown in Figure 2.

The pulsed $I_{DS}-V_{DS}$ measured data of the device under different gate quiescent bias voltages $V_{gsq}$ are shown in Figure 3a, and the pulsed $I_{DS}-V_{DS}$ measured data under different drain quiescent bias voltages $V_{dsq}$ are shown in Figure 3b. The pulse width is 1 μs and the pulse duty cycle is 0.1%. As can be seen from Figure 3, $R_{on}$ increases when $V_{gsq}$ decreases and $V_{dsq}$ increases. The drain current $I_{ds}$ decreases initially but slowly recovers later when $V_{ds}$ increases. The maximum drain current $I_{ds\_max}$ decreases with the decrease in $V_{gsq}$ and the increase in $V_{dsq}$. By observing Figure 3, we also find that the values of model parameters, such as $V_{off}$ and $\mu_{a}$, change with $V_{gsq}$ and $V_{dsq}$, that is, different quiescent bias voltages may bring distinct extents of trapping effects. For a given temperature, the drain current $I_{ds}$ is a function of $V_{gs}$, $V_{ds}$, $V_{gsq}$, and $V_{dsq}$.

Since the model parameters $V_{off}$, $\mu_{a}$, $R_{drain}$, and DIBL-effect-related parameter ($V_{dscale}$) change with the quiescent bias voltages $V_{gsq}$ and $V_{dsq}$, the trapping effect is added to these model parameters affected by the trapping effect through Equation (5), capturing the trend of the parameters changing with $V_{gsq}$ and $V_{dsq}$. $V_{g\_trap}$ and $V_{d\_trap}$ are trap voltages generated by the trap networks, *P* is the parameter value after adding the trapping effect, $P_{ref}$ is the initial parameter value when $V_{gsq}$ and $V_{dsq}$ are 0, and the trP0 and trP1 are fitting parameters.
(5)$$P=P_{ref}+trP0\cdot V_{g\_trap}+trP1\cdot V_{d\_trap}$$

In order to extract the variation coefficients trP0 and trP1 of the model parameters, it is necessary to fit the pulsed $I_{DS}-V_{DS}$ measured data of GaN HEMTs under different quiescent bias voltages $V_{gsq}$ and $V_{dsq}$. The value of trP0 can be extracted through matching features when we treat $V_{dsq}$ as a constant value (i.e., 0 V) but sweep the value of $V_{gsq}$. Similarly, the value of trP1 can be approximated by maintaining the value of $V_{gsq}$.

### 2.2. Modeling the Effects of Threading Dislocations on Gate Current

The gate leakage current in AlGaN/GaN HEMTs at room temperature can be modeled as follows:(6)$$I=Area\cdot \left(J_{TE}+J_{PF}\right)$$
where Area is the area of the gate, $J_{TE}$ is the thermionic emission (*TE*) current density, and $J_{PF}$ is the *PF* emission current density. *PF* emission is considered to be the main leakage mechanism for gate current conduction in the low-to-medium reverse-bias region [25,26].

First-principles calculations indicate that pure threading dislocations introduce deep occupied states both above the valence-band maximum (VBM) and below the conduction-band minimum (CBM) of GaN, resulting in the pure dislocation state and the V_III_-decorated dislocation state [32]. Since the PF emission originates from the transport of electrons via the continuum dislocation state [15], the two-state FP emission should correspond to two distinct dislocation states in the AlGaN barrier, which are the pure dislocation state and the V_III_-decorated dislocation state. The typical conduction band diagram for medium reverse gate voltage, which describes this mechanism, is shown in Figure 4, where the continuums of states are at heights $\Phi_{d1}$ and $\Phi_{d2}$ from the Schottky metal. The trap states in the barrier are assumed to be very close to the metal Fermi level. As depicted in Figure 4a, the electrons firstly transfer along the low-energy V_III_-decorated dislocation level. As the reverse voltage increases, electrons in the conducting V_III_-decorated dislocation state become saturated and begin to transfer along the pure dislocation state when the excess electrons gain sufficient energy. The two-state PF emission mechanism plays a dominant role at low-to-medium reverse bias.

For AlGaN/GaN heterojunction, we present a new transport mechanism of two-state PF emission to model the gate leakage current under reverse voltage with the effects of threading dislocations.

The relation between the current density $J_{PF}$ and the electric field (E) for PF conduction is given by [33]:(7)$$J_{PF\_n}=CEexp\left[-\frac{q\left(\Phi_{dn}-\sqrt{qE/\pi \varepsilon_{0}\varepsilon_{s}}\right)}{k_{B}T}\right]$$
(8)$$E=\frac{q\sigma_{p}-C_{g}\left(V_{G}-\varphi_{GaN}\right)}{\varepsilon_{s}}$$
(9)$$J_{PF}=J_{PF\_1}+J_{PF\_2}$$
where C is a parameter dependent on the trap concentration and *E* is the electric field in the AlGaN barrier at the metal–semiconductor interface. $\Phi_{dn}$ (i.e., $\Phi_{d1}=$ 0.47 eV and $\Phi_{d2}=$ 1.07 eV) is the barrier height for the electron emission from the Schottky metal to the dislocation states of the AlGaN barrier, $\varepsilon_{0}$ is the permittivity of free space, $\varepsilon_{s}$ is the relative dielectric permittivity at high frequency, T is the temperature, q is the electron charge, and $k_{B}$ is the Boltzmann constant. Additionally, $\sigma_{p}$ is the sum of the piezoelectric polarization charge in the barrier and the difference between spontaneous polarization charge in the barrier and the buffer, and $C_{g}$ is the gate capacitance.

The parameters related to the proposed gate leakage current model with effects of threading dislocations are C and $\sigma_{p}$. We extract the parameters of a particular current component in the region where it is dominant. At low-to-medium reverse bias, PF emission current is dominant. The values of C and $\sigma_{p}$ can be obtained from the slope and intercept of the plot of $\ln\left(J_{PF}/E\right)$ versus $V_{G}$.

## 3. Model Validation and Analysis

The device studied in this paper is a commercial GaN HEMT device with a gate length of 0.25 μm and a total gate width of 6 × 125 μm. To validate the trap model that captures the trapping effects on drain current, we varied $V_{dsq}$ and $V_{gsq}$ and obtained the pulsed output characteristics, that is, 1 μs pulse width and 0.1% pulse duty cycle. The trap model proposed in this paper is used for simulation and parameter extraction. Table 1 shows the extracted values of the model parameters of the proposed trap model. The model parameters, which are affected by the trapping effects, are the $V_{off}$, $\mu_{a}$, $R_{drain}$, and $V_{dscale}$. The extracted values of the coefficients trP0 and trP1 are shown in Table 2.

Figure 5 shows the pulsed $I_{DS}-V_{DS}$ measured data and modeled data under different quiescent bias voltages using the trap model proposed in this paper. In this figure, the lines represent the modeled data while the symbols refer to the measured data. As shown in Figure 5, there is an excellent match between the model data and measured data under different quiescent bias voltages, which demonstrates that the proposed trap model can accurately characterize the trapping effects caused by threading dislocations on drain current of GaN HEMTs.

Figure 6 depicts the measured and simulated curves of the gate current of an AlGaN/GaN HEMT as a function of bias voltage at 300 K. Form low to medium reverse bias voltages, PF emission current dominates at room temperature conditions. As one can see, two distinct regions (regions I and II) have different voltage dependencies of the current, which indicates the existence of two distinct PF emission states. As demonstrated in Figure 6, the simulated gate current fits well with the measured data. At present, we only have measured data at 300 K temperature, and the verification at other temperatures will be carried out later.

## 4. Conclusions

In this paper, we model the effects of threading dislocations on both gate current and drain current of AlGaN/GaN HEMTs. The RC subcircuits with diodes can accurately reflect the dynamic characteristics of charge trapping and detrapping in GaN HEMTs. Meanwhile, we propose a new transport mechanism, the two-state PF emissions resulting from two distinct electron-transfer channels, which exhibit voltage dependence on the gate current. The simulation results of drain current and gate current fit well with the measured data. This model was implemented in Verilog-A and simulations were performed with the help of Keysight ICCAP. It is ready to be deployed in our SP-based GaN HEMT model [34] which is an important step toward developing a complete compact model for GaN devices.

## Figures and Tables

**Figure 1 micromachines-14-00305-f001:**
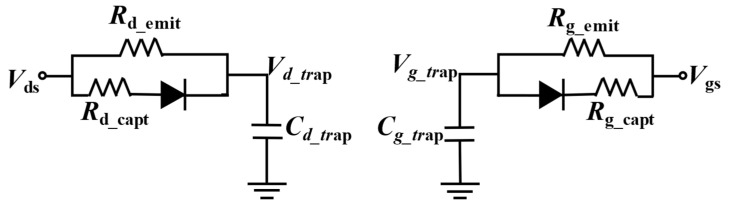
Equivalent circuit topology of the trap model of drain current proposed in this paper.

**Figure 2 micromachines-14-00305-f002:**
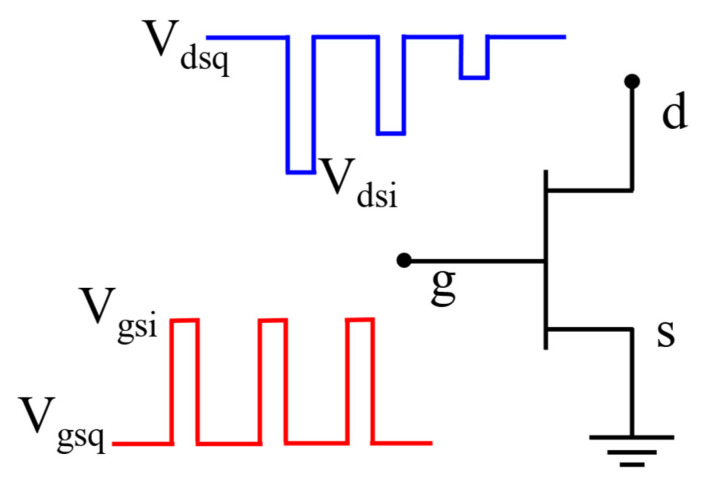
Signal settings for pulsed $I_{DS}-V_{DS}$ measurement.

**Figure 3 micromachines-14-00305-f003:**
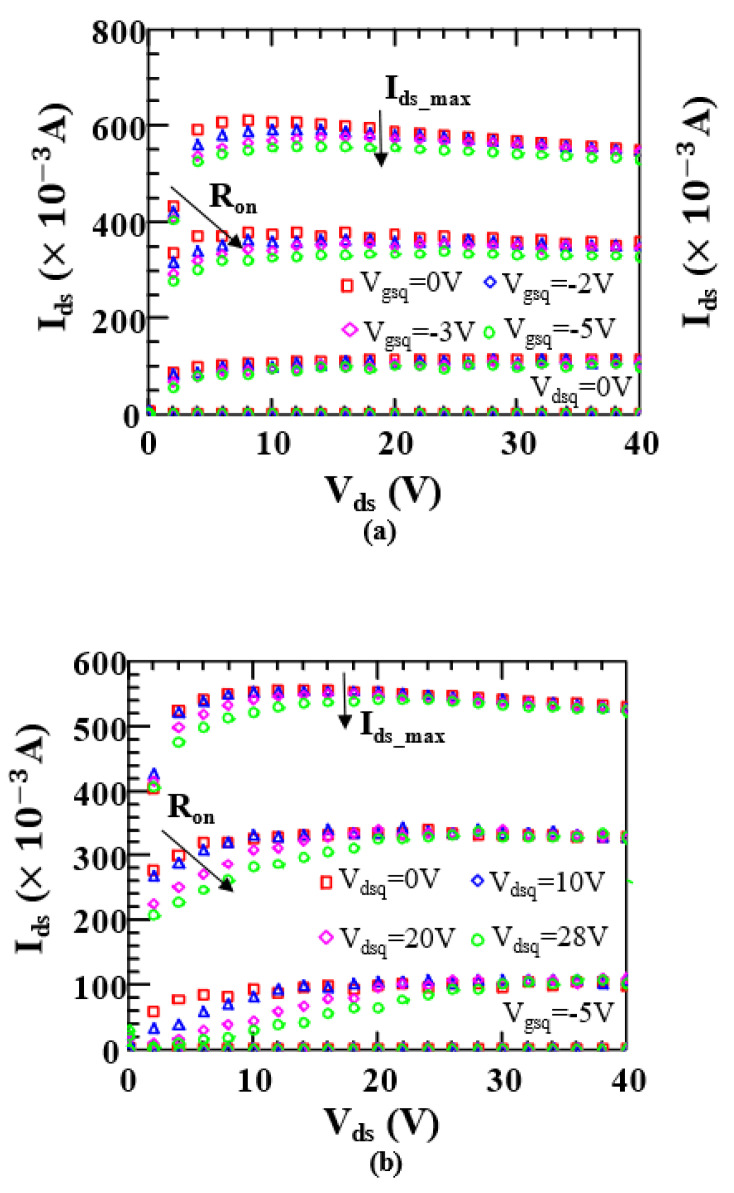
Measured pulsed output characteristics under (**a**) different gate quiescent bias voltages $V_{gsq}$ and (**b**) different drain quiescent bias voltages $V_{dsq}$. The gate voltage $V_{gs}$ ranges from −3 V to 0 V with a step size of 1 V.

**Figure 4 micromachines-14-00305-f004:**
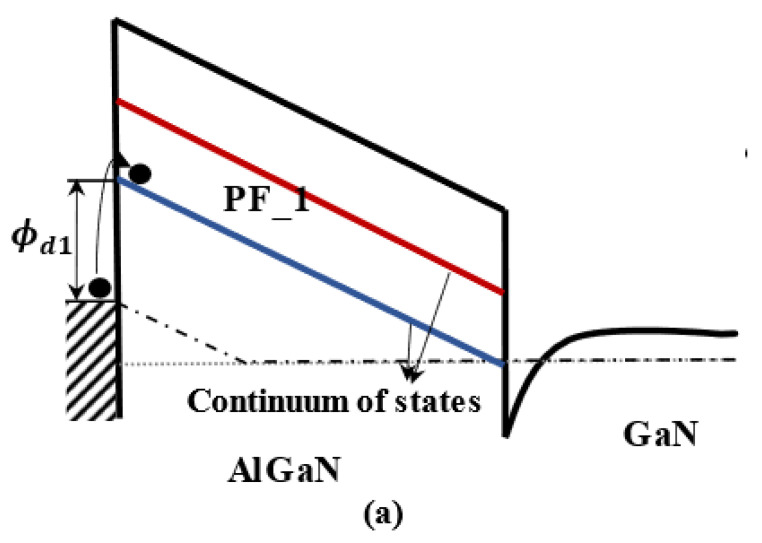
Conduction band edge diagrams of AlGaN/GaN HEMT under reverse gate voltage showing PF emissions via (**a**) a V_III_-decorated dislocation state and (**b**) both a V_III_-decorated dislocation state and a pure dislocation state.

**Figure 5 micromachines-14-00305-f005:**
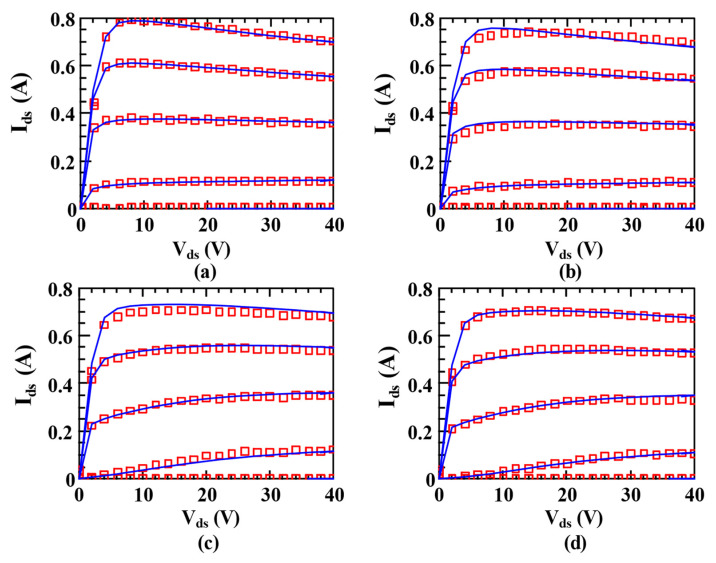
Measured (symbol) pulsed $I_{DS}-V_{DS}$ data and modeled (line) pulsed $I_{DS}-V_{DS}$ data using the trap model proposed in this paper under different quiescent bias at 25 °C: (**a**) $V_{gsq}$ = 0 V, $V_{dsq}$ = 0 V; (**b**) $V_{gsq}$ = −3 V, $V_{dsq}$ = 0 V; (**c**) $V_{gsq}$ = −3 V, $V_{dsq}$ = 28 V; (**d**) $V_{gsq}$ = −5 V, $V_{dsq}$ = 28 V. $V_{gs}$ ranges from −4 V to 1 V with a step size of 1 V.

**Figure 6 micromachines-14-00305-f006:**
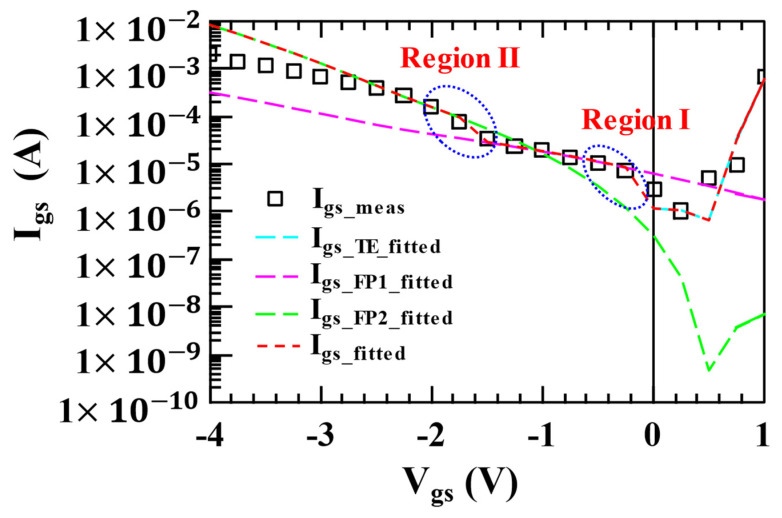
Measured and simulated curves of the gate current of an AlGaN/GaN HEMT as a function of bias voltage at 300 K.

**Table 1 micromachines-14-00305-t001:** Parameters of trap model.

Parameters	Values
$R_{g\_emit}\left[\times 10^{3}\;\Omega \right]$	100
$R_{g\_capt}\left[\Omega \right]$	2
$C_{g\_trap}\left[\times 10^{-12}F\right]$	200
$R_{d\_emit}[\times 10^{3}\;\Omega$]	120
$R_{d\_capt}\left[\Omega \right]$	2
$C_{d\_trap}\left[\times 10^{-12}F\right]$	200

**Table 2 micromachines-14-00305-t002:** Extracted values of coefficients of parameters varying with trap voltages.

Parameters	trP0	trP1
$V_{off}$	−40.42 m	26.67 m
$\mu_{a}$	−1.197 n	90 p
$R_{drain}$	−100 m	10 m
$V_{dscale}$	−991.2 m	1.413

## Data Availability

The data presented in this study are available on request from the corresponding author.

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
