# Peer review of "Modeling the Effects of Threading Dislocations on Current in AlGaN/GaN HEMT"

_micromachines, 2023, doi:10.3390/mi14020305_

Round 1
Reviewer 1 Report
In this paper, the authors have modeled the effects of threading dislocations on AlGaN/GaN HEMTs. They also modeled the dynamic of charge trapping and de-trapping by using RC subcircuits with diodes and presented a novel Poole-Frenkel (PF) model to describe the gate leakage current. However, the reviewer has a few questions about the technical content.
1. On page 3, line 111, the authors mention that the emission time of the trap is about 1e-3 seconds. However, in Table 1 and reference 30, it seems that the emission time should be about 1e-5 seconds. Please clarify.
2. On page 4, Figure 3, the changes of Ron under different Vgsq and Vdsq should be added either in figures or texts to demonstrate the changes clearly.
3. On page 4, Figures 3(a) and 3(b) show that both gate-lag and drain-lag will cause current collapse. However, it seems that the current of the drain-lag measurement will recover to the initial fresh value with the increase of drain voltage, but the gate-lag measurement does not show the same condition. The authors are advised to explain if there is any different mechanism of trapping or the type of traps that will cause these results.
4. There are many inconsistencies between “trp1” and “trP1.” Please double-check and use the same notation. E.g., on page 4, line 139, and page 6, line 195.
5. Page 4, Figure 4 shows that the level difference between two PF emissions states and Ec should not change as reverse voltage increases.
6. On page 6, Table 1, there is a lack of a right bracket: “??_???? [× 103Ω” is advised to be corrected to “??_???? [× 103Ω].”
7. There are some grammar problems. The authors are advised to seek professional editing services.
8. The references of parameters used in Table 1 should be provided. Also, the unit should be included.
Author Response
Dear reviewer:
Thank you for your decision and constructive comments on my manuscript. We have carefully considered the suggestion of Reviewer and make some changes. We have tried our best to improve and made some changes in the manuscript.
Revision notes, point-to-point, are given in the attachment. We hope you find our modifications acceptable for peer-review, but please let me know otherwise. We would be happy to further revise the manuscript if this is not to your satisfaction.
Many thanks,
Wang, Jie
Zhejiang Provincial Key Laboratory of Large-scale Integrated Circuit Design
Hangzhou Dianzi University
Hangzhou 310018, China
wangjie@hdu.edu.cn

Reviewer 2 Report
General comment: The overall intention of this manuscript is a good one. I agree that more attention should be paid to the modeling and understanding the effects of threading dislocations on both gate and drain currents of GaN based HEMTs. The model used in this submission is simple and easy to understand: Two parallel RC subcircuits with diodes are used to characterize the trapping effects on drain current. The trapping effects are modeled by modifying the model parameters such as cut-off voltage according to trap voltages generated by RC subcircuits to capture the drain current drop caused by traps.
After a careful reading of this submission these are my remarks for the Editor/Author to consider:
Minor suggestions
Minor comment 1: Consider shortening and rephrasing the sentences in the entire manuscript.
Minor comment 2: In general, the presentation of the tables and figures in this submission requires improvement throughout.
Minor comment 3: Some references that are more than 20 years old should be replaced with more recent ones.
Major suggestions
Major comment 1: I would consider improving the abstract and conclusion of this submission.
Major comment 2: In the introduction of this manuscript the authors aim to propose a helpful figure to understand the general framework.
Major comment 3: The paper written requires substantial revision. Specifically, several sentences are long and hard to read or understand.
Major comment 4: In the Introduction, the way the authors present the interventions of other studies in relation to the subject of this submission is very weak and need serious revision. Comparisons between these different interventions are well requested in order to improve the quality of the introduction.
Major comment 5: The authors should include a brief motivation and novelty section in his manuscript to explain the significance of his research compared with the available findings in the literature?
Major comment 6: The physical explanations of the crucial findings in this submission require a deep improvement.
Major comment 7: The authors should include some details and explain the performance of EDA Software used in his paper for simulation.
Major comment 8: I will be glad if the authors confirm their results by using the available ones in the literature.
Author Response

(The authors gave the same response as above.)

Round 2
Reviewer 1 Report
In this paper, the authors have modeled the effects of threading dislocations on AlGaN/GaN HEMTs. They also modeled the dynamic of charge trapping and de-trapping using RC subcircuits with diodes and presented a novel Poole-Frenkel (PF) model to describe the gate leakage current. However, the reviewer has a few questions about the technical content.
1. On page 3, line 111, the authors mention that the emission time of the trap is about 1e-3 seconds. However, in Table 1 and reference 30, it seems that the emission time should be about 1e-5 seconds. Please clarify.
2. On page 4, Figure 3, the changes of Ron under different Vgsq and Vdsq should be added either in figures or texts to demonstrate the changes clearly.
3. On page 4, Figures 3(a) and 3(b) show that both gate-lag and drain-lag will cause current collapse. However, it seems that the current of the drain-lag measurement will recover to the initial fresh value with the increase of drain voltage, but the gate-lag measurement does not show the same condition. The authors are advised to explain if there is any different trapping mechanisms or the type of traps that will cause these results.
4. There are many inconsistencies between “trp1” and “trP1.” Please double-check and use the same notation. E.g., on page 4, line 139, and page 6, line 195.
5. Figure 4 shows the difference between two PF emissions states, and Ec should not change as reverse voltage increases.
6. On page 6, Table 1, there is a lack of a right bracket: “??_???? [× 103Ω” is advised to be corrected to “??_???? [× 103Ω].”
7. There are some grammar problems. The authors are advised to seek professional editing services.
8. The references of parameters used in Table 1 should be provided. Also, the unit should be included.
Author Response
Dear reviewer:
Thank you for your decision and constructive comments on my manuscript. We have carefully considered the suggestion of Reviewer and make some changes. We have tried our best to improve and made some changes in the manuscript.
Most of the problems were solved last time. This time we improved the English writing of the manuscript. We hope you find our modifications acceptable for peer-review, but please let me know otherwise. We would be happy to further revise the manuscript if this is not to your satisfaction.
Many thanks,
Wang, Jie
Zhejiang Key Laboratory of Large-scale Integrated Circuit Design
Hangzhou Dianzi University
Hangzhou 310018, China
wangjie@hdu.edu.cn
Reviewer 2 Report
The Submission has been well improved and the revision is clearly presented. I think most of my suggestions and comments have been addressed.
In my opinion this paper is worth to be published in Micromachines Journal.
Author Response
Dear reviewer:
Thank you for your decision on my manuscript. We are very happy that you find our modifications acceptable for peer-review. Thanks again for your review and help.
Many thanks,
Wang, Jie
Zhejiang Key Laboratory of Large-scale Integrated Circuit Design
Hangzhou Dianzi University
Hangzhou 310018, China
wangjie@hdu.edu.cn